# Comparative Transcriptome Identifies Gene Expression Networks Regulating Developmental Pollen Abortion in Ogura Cytoplasmic Male Sterility in Chinese Cabbage (*Brassica rapa* ssp. *pekinensis*)

Lijiao Hu [1,2,†], Xiaowei Zhang [1,†], Yuxiang Yuan [1,†], Zhiyong Wang [1], Shuangjuan Yang [1], Ruina Li [2], Ujjal Kumar Nath [3], Yanyan Zhao [1], Baoming Tian [2], Gongyao Shi [2], Zhengqing Xie [2], Fang Wei [1,2,*] and Xiaochun Wei [1,2,*]

[1] Institute of Horticulture, Henan Academy of Agricultural Sciences, Graduate T&R Base of Zhengzhou University, Zhengzhou 450002, China; hu_lijiao@163.com (L.H.); xiaowei5737@163.com (X.Z.); yuxiangyuan126@126.com (Y.Y.); nkywzy@163.com (Z.W.); sjyang_0614@163.com (S.Y.); zhaoyanyan9621@163.com (Y.Z.)

[2] Henan International Joint Laboratory of Crop Gene Resources and Improvement, School of Agricultural Sciences, Zhengzhou University, Zhengzhou 450001, China; ruinali0404@163.com (R.L.); tianbm@zzu.edu.cn (B.T.); shigy@zzu.edu.cn (G.S.); zqxie@zzu.edu.cn (Z.X.)

[3] Department of Genetics and Plant Breeding, Bangladesh Agricultural University, Mymensingh 2202, Bangladesh; ujjalnath@gmail.com

[*] Correspondence: fangwei@zzu.edu.cn (F.W.); jweixiaochun@126.com (X.W.); Tel.: +86-371-6778-5055 (F.W.); +86-371-6571-4026 (X.W.)

[†] Equally contribution.

**Abstract:** Ogura cytoplasmic male sterility (Ogura CMS), originally identified in wild radish (*Raphanus sativus*), has enabled complete pollen sterility in *Brassica* plants, but the underlying mechanism in Ogura CMS Chinese cabbage (*Brassica rapa* ssp. *pekinensis*) remains unclear. In this study cytological analysis showed that during microsporogenesis the meiosis occurred normally, and the uninucleated pollens subsequently formed, but the development of both binucleated and trinucleated pollens was obviously disrupted due to defects of pollen mitosis in the Ogura CMS line (Tyms) compared with the corresponding maintainer line (231–330). In transcriptome profiling a total of 8052 differentially expressed genes (DEGs) were identified, among which 3890 were up-regulated and 4162 were down-regulated at the pollen abortion stages in an Ogura CMS line. KOG cluster analysis demonstrated that a large number of DEGs were related to the cytoskeleton's dynamics, which may account for the failure of pollen mitosis during development in the Ogura CMS line. The pivotal genes related to the phenylpropane synthesis pathway (*PAL*, *4CL* and *CAD*) were significantly down-regulated, which probably affected the formation and disposition of anther lignin and sporopollenin, and eventually led to abnormality in the pollen exine structure. In addition, several key up-regulated genes (*GPX7*, *G6PD* and *PGD1*) related to the glutathione oxidation-reduction (REDOX) reaction indicated that the accumulation of peroxides in Ogura CMS lines during this period affected the pollen development. Taken together, this cytological and molecular evidence is expected to advance our understanding of pollen abortion induced by Ogura cytoplasmic action in Chinese cabbage.

**Keywords:** Ogura cytoplasmic male sterility; pollen abortion; transcriptome; phenylpropane synthesis; Chinese cabbage

## 1. Introduction

Male sterility refers to a failure to develop normal anthers or pollens, but the plants may still develop normal pistils [1]. Both nuclear and cytoplasmic genes are generally involved in regulating male sterility. There are two main types of male sterility: genic male sterility (GMS), governed by nuclear gene(s), and cytoplasmic male sterility (CMS), regulated by the interaction of nuclear and cytoplasmic gene(s) [2]. Specifically, CMS

found in many higher plants is governed by maternal inheritance with aborted pollens and normal pistils [3]. CMS plants can manifest pollen abortion depending on the flower apparatus, short filaments and thin anthers in morphology [4]. CMS has been reported in more than 150 plant species to date [5].

In *Brassica*, CMS is determined by the mitochondrial genome and associated with a pollen-sterility phenotype that can be suppressed or counteracted by nuclear genes known as fertility-restorer genes [6]. Ogura CMS was first observed in Japanese radish (*Raphanus sativus*) and is now widely used in breeding *Brassica* crops [7]. The pollen abortion in Ogura CMS cabbage mostly occurred at microspore developmental stages due to the disintegration of the plasma membrane and an abnormal tapetum, which provides nutrition for microspores [8,9]. In microsporogenesis, tapetum cells secrete callose enzymes after the completion of the meiosis of microspore mother cells, which break down the callose wall and release microspores. By contrast, the premature disintegration or delayed degradation of tapetum cells may result in pollen abortion [10].

Ogura CMS is controlled by a mitochondrial *ORF138*, which consists of two co-transcribed open reading frames *ORF138* and *ATP synthase subunit8* (*ATP8*), but the changes in a series of other genetic backgrounds and molecular mechanisms caused by the *orf138* mutation are still unclear [11]. *ATPase* catalyzes the hydrolysis of ATP to provide the energy necessary for pollen development [12]. With the development of pollens, the activity and quantity of *ATPase* theoretically increase, otherwise the pollens will be abortive. Calcium ions act as a second messenger pathway for signal transduction during the development of microspores and gametophytes [13]. In addition, the cytoskeletal organization, programmed cell death, hormone balance and peroxide content may also influence the degrees of pollen fertility [14–16].

Genes involved in carbon metabolism, lipid metabolism, the tricarboxylic acid (TCA) cycle and oxidative phosphorylation are critical for pollen fertility and microspore development in Ogura CMS cabbage lines [17]. These genes could regulate pollen development through changes their expression either at an early stage of pollen development or at the maturity of pollen. In *B. rapa*, Ogura CMS flower buds had lower contents of soluble protein, soluble sugar, free proline, catalase (CAT), peroxidase (POD) and superoxide dismutase (SOD) than maintainer lines [18]. However, the molecular mechanism associated with Ogura CMS, particularly its core molecular mechanism, remains unclear. Therefore, the elucidation of the molecular mechanism of pollen abortion will provide a theoretical basis for the further understanding of the mechanism of male sterility in plants.

In recent years, transcriptome sequencing has been widely used in soybeans [19], cotton [20], rice [21] and other crops as a powerful tool for studying global transcriptional networks to provide high-resolution data, such as those on leaf senescence, leaf color and biological and abiotic stress responses [22]. However, the use of the transcriptome analysis of pollen abortion and male sterility in Chinese cabbage is very limited. In plants, the anther and pollen development process is very complicated, involving gene expression, regulation, metabolism and the activity of many genes. As long as one part of these processes is disturbed, pollen development may be blocked, and pollen abortion may occur [23]. Therefore, high-throughput sequencing methods can help us to better obtain key information regarding pollen development.

In the present study, transcriptome sequencing was performed using stamens during microsporogenesis between Ogura CMS and its maintainer line of Chinese cabbage, to identify critical DEGs related to pollen development. Our findings will help to elucidate the mechanisms controlling sterility at different stages of pollen development in the Ogura CMS system.

## 2. Results

### 2.1. The Ogura CMS Chinese Cabbage Displays Complete Male Sterility

Morphologically, the Ogura CMS Chinese cabbage line (Tyms) had a similar flower pattern to its maintainer line (231–330), but the organs (stamens, petals and sepals) were

mostly smaller in the Ogura CMS line (Figure S1 and Table S1). At the mature stage, the anthers were completely non-dehiscent and empty in the Ogura CMS line, but the anthers in the maintainer line were naturally split and full of pollen grains (Figure 1). Compared with the maintainer line, the meiosis was normal in the Ogura CMS line (Figure S2), and tetrads were formed after the completion of meiosis. However, the uninucleated pollens could normally form tetrads, but were unable to develop further into binucleated and trinucleated pollens in the Ogura CMS line (Figure 2), which was probably due to the defective mitosis during pollen development.

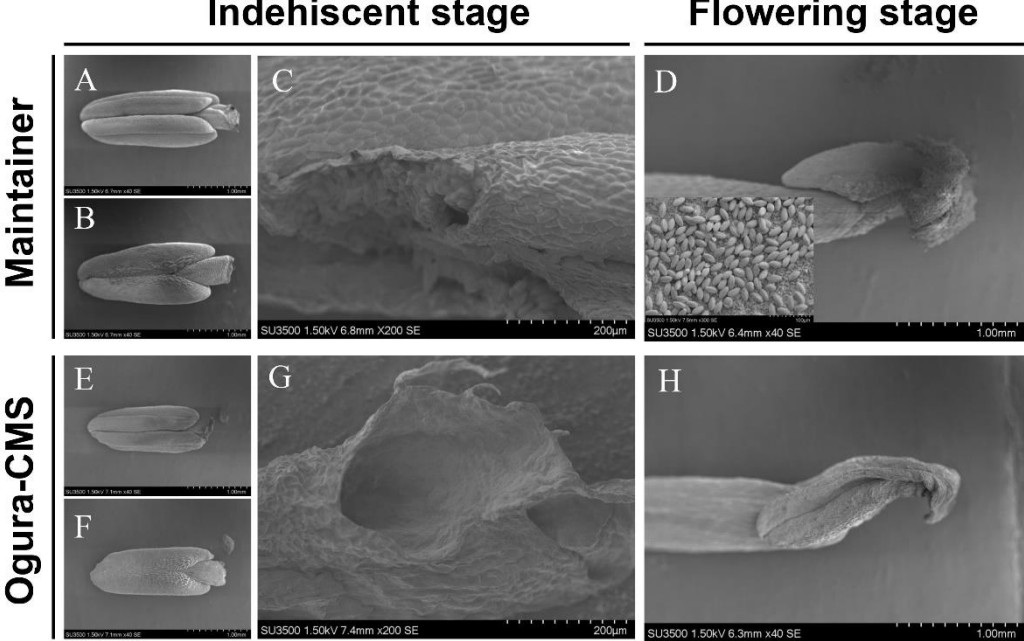

**Figure 1.** Scanning electron microscopy observation of anthers in the Ogura CMS line and the maintainer line of Chinese cabbage. (**A**,**B**) mature anthers at the indehiscent stage from the maintainer line (231–330); (**C**) local magnification of a mature anther incised at the indehiscent stage in the maintainer line; (**D**) mature anther at the flowering stage with normal oval, plump pollen grains in the maintainer line; (**E**,**F**) Mature anthers at the indehiscent stage from the Ogura CMS line (Tyms); (**G**) local magnification of a mature anther incised at the indehiscent stage in the sterile line, showing empty chambers; (**H**) a mature anther at the flowering stage had no pollen and was collapsed in the sterile line. Bar = 1 mm in (**A**,**B**,**E**,**F**), 200 μm in (**C**,**G**), and 1 mm in (**D**,**H**).

*2.2. Posttranscriptional Regulation, Carbohydrate Metabolism and Cytoskeleton Dynamics Were Probably Associated with Pollen Abortion in Ogura CMS Line*

For comparison with the maintainer line, the young anthers with developing pollens of the Ogura CMS line were collected and subjected to RNA sequencing. Differentially expressed genes (DEGs) were identified with the criteria of fold change (FC) $\geq 2$ and false discovery rate (FDR) <0.05. A total of 8052 genes differentially expressed between the maintainer and Ogura CMS lines were identified. Among them, 3890 and 4162 showed up- and down-regulation, respectively (Figure 3). The identified DEGs were classified into 25 categories by using the KOG database (Figure 4A and Table S2). Except for the term "transcription", the category "posttranslational modification, protein turnover, chaperones" was comparatively highlighted, which indicated that posttranscriptional regulation might play important roles in modulating pollen sterility during development in the Ogura CMS line. Additionally, 326 DEGs were also categorized into the term "carbohydrate transport and metabolism" indicating a close relationship between energy metabolism and pollen fertility. In addition, the DEGs were partially classified into two related categories termed "cell cycle control, division, chromosome partitioning" and "cytoskeleton", which likely

demonstrates that pollen mitosis and cytoskeletal dynamics were associated with the binucleated and trinucleated pollen development after meiotic tetrad formation. Intact cytoskeleton was essential for pollen growth, and insufficient expression of *ACT* and activity may lead to pollen abortion [24]. For instance, a number of *ACTIN* genes including *ACT1* (*Bra005178* (log$_2$FC: $-0.969$); *Bra000010* (log$_2$FC: $-7.701$); *Bra017166* (log$_2$FC: $-2.730$)), *ACT3* (*Bra014865* (log$_2$FC: $-1.820$); *Bra007025* (log$_2$FC: $-1.515$)), *ACT4* (*Bra020319* (log$_2$FC: $-10.268$)) and *ACT12* (*Bra033819* (log$_2$FC: $-8.022$); *Bra019486* (log$_2$FC: $-9.227$); *Bra018210* (log$_2$FC: $-9.654$)) were obviously down-regulated in the Ogura CMS line (Figure 4B and Table S3). Among them, *ACT4* and *ACT12* genes have been shown to be mainly expressed in Arabidopsis anthers by GUS staining [25]. In addition, the actin depolymerizing factor (ADF), an important player in actin remodeling, increases actin filament treadmilling rates. Furthermore, it has been reported that the actin filaments of *ADF7* and *ADF10* mutants were reduced in the shank and tip, respectively [26]. Another gene, *VLN5*, a typical member of the *VILLIN* family affecting pollen tube growth, were also negatively affected in the Ogura CMS line [27].

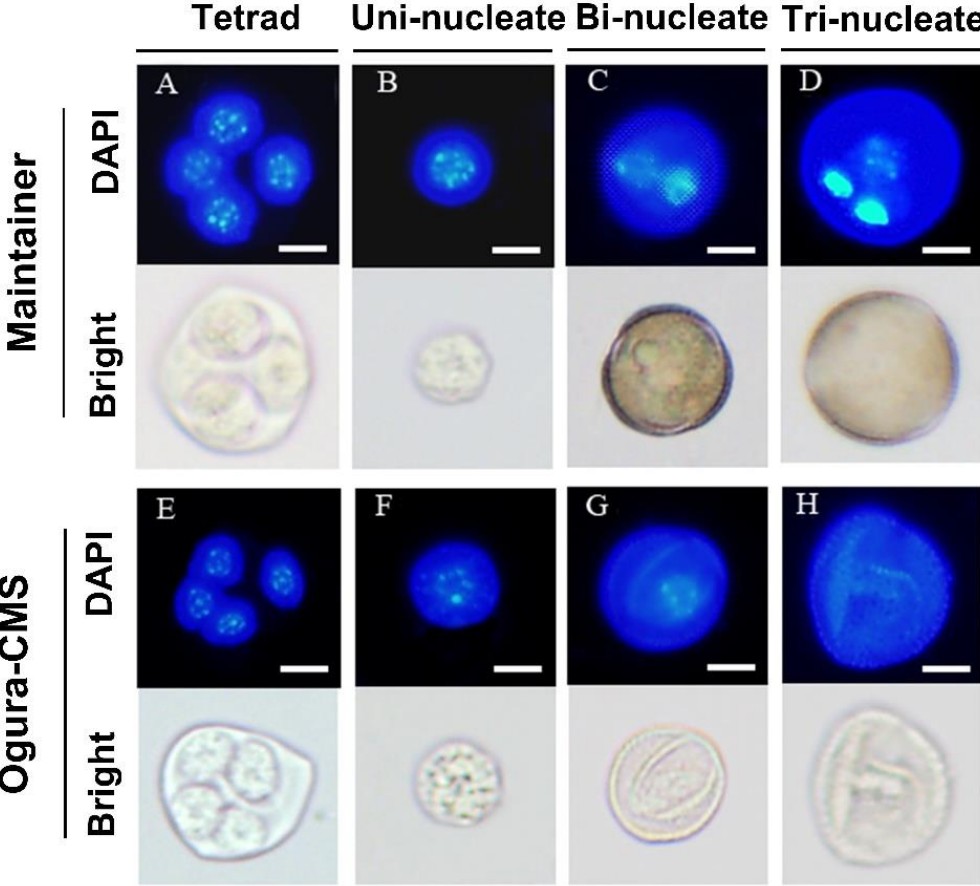

**Figure 2.** DAPI staining of microspore development process of the Ogura CMS line (Tyms) and the maintainer line (231–330) in Chinese cabbage. (**A–D**) DAPI-stained developing spores from the maintainer line (231–330); (**E–H**) DAPI-stained developing spores from the Ogura CMS line (Tyms). (**A,E**): Tetrad; (**B,F**): Uni-nucleate; (**C,G**): Bi-nucleate; (**D,H**): Tri-nucleate; Bar = 50 μm.

## 2.3. Up-Regulated Expression of GSH-Oxidation Genes Probably Led to ROS Accumulation and Affected Pollen Fertility in the Ogura CMS Line

In the GO annotations, which include biological process, cellular component and molecular function, the up-regulated DEGs in the top five categories were mostly enriched with "glutathione binding" (GO:0043295) and "glutathione transferase activity" (GO:0004364) (Figure 5A). Glutathione was considered an important metabolite regulator

participating in the tricarboxylic acid (TCA) cycle and sugar metabolism and promoting carbohydrate and fat metabolism [28]. In the glutathione oxidation-reduction (REDOX) reaction, the reduced glutathione (GSH) and oxidized glutathione (GSSG) could have transformed each other in the presence of glutathione peroxidase (GPX), glutathione reductase and nicotinamide adenine dinucleotide phosphate (NADPH), accompanied by the decomposition of hydrogen peroxide [29]. Therefore, compared with the maintainer line, the up-regulation of *GPX* genes—*GPX6* (*Bra035211* (log$_2$FC: 1.083)) and *GPX7* (*Bra023978* (log$_2$FC: 1.568)) and glucose 6-phosphate dehydrogenase genes—*PGD1* (*Bra027685* (log$_2$FC: 2.108)) and *G6PD2* (*Bra006181* (log$_2$FC: 2.059); *Bra008855* (log$_2$FC: 1.110)) during pollen development in the Ogura CMS line suggested a persistent GSH-to-GSSG transformation and accumulation of reactive oxygen species (ROS) such as superoxide, which might eventually affect pollen sterility (Figure 5B).

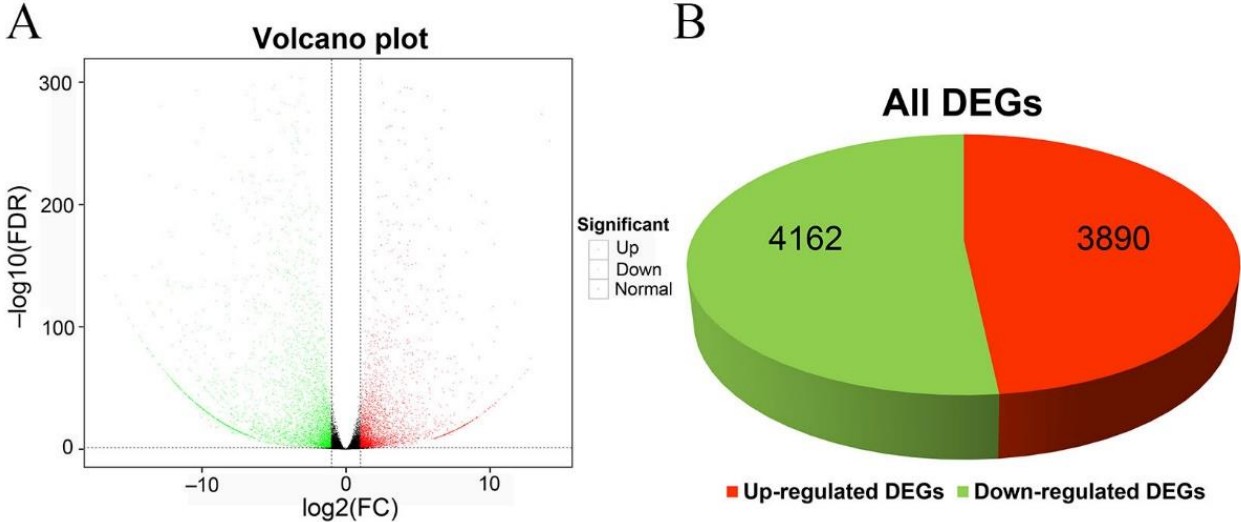

**Figure 3.** Differentially expressed genes (DEGs) in the Ogura CMS (Tyms) and its maintainer (231–330) lines; (**A**) volcano plot involving DEGs; (**B**) number of DEGs.

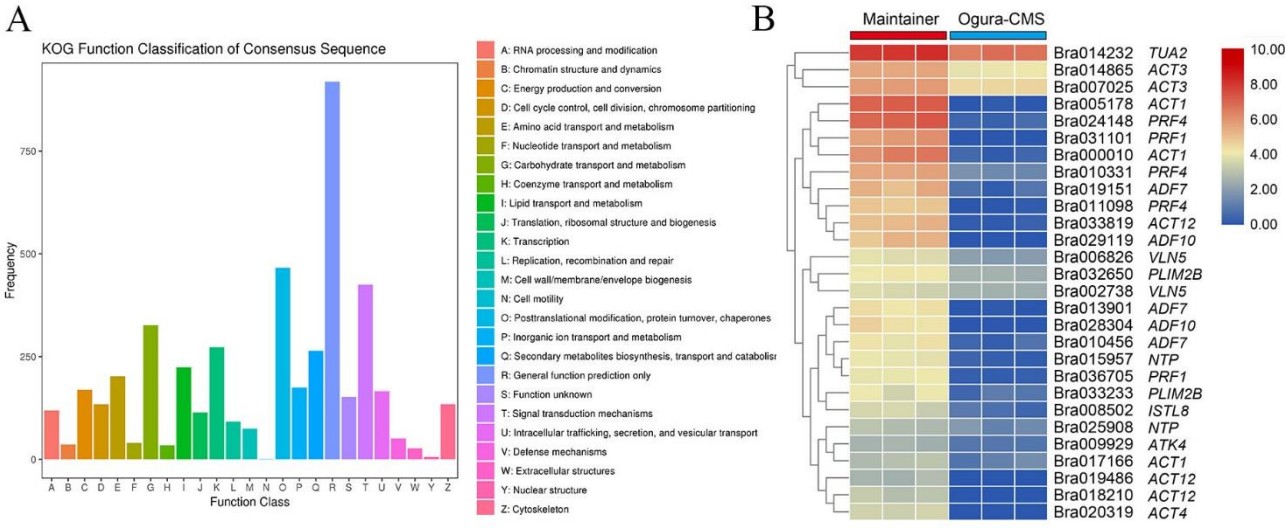

**Figure 4.** The differentially expressed genes by KOG enrichment in the Ogura CMS (Tyms) and its maintainer (231–330) lines; (**A**) KOG clusters involving DEGs; (**B**) heatmap of DEGs that are related to the cytoskeleton (Z) according to KOG enrichment analysis.

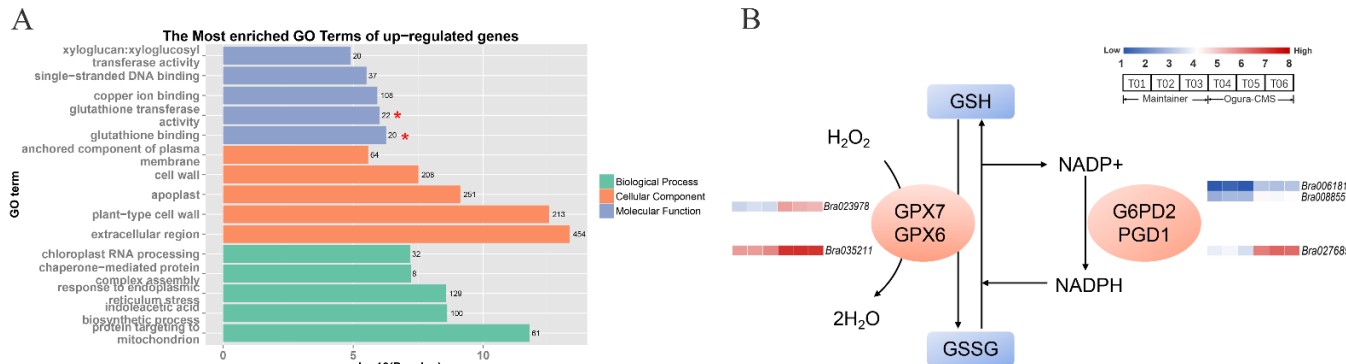

**Figure 5.** Enrichment analysis of genes up-regulated in the maintainer line vs. Ogura CMS line. (**A**) GO analysis of up-regulated DEGs; (**B**) part of the glutathione metabolic pathway. GSH, glutathione; GSSG, oxidized glutathione; G6PD2, glucose-6-phosphate dehydrogenase 2; PGD1, 6-phosphogluconate dehydrogenase 1; GPX7, glutathione peroxidase 7; GPX6, glutathione peroxidase 6. T01, T02 and T03 represented 3 libraries (the maintainer line, 3 times replication); T04, T05 and T06 represented 3 libraries (the Ogura CMS line, 3 times replication).

### 2.4. Down-Regulation of DEGs Related to Phenylpropane Synthesis May Affect Sporopollenin Formation during Pollen Development in the Ogura CMS Line

Phenylpropane is an important secondary metabolite, and the pathway of phenylpropane synthesis is related to plant-cell-wall exine during pollen development [30]. Coincidentally, we observed that identified DEGs were classified into the GO term "pollen exine formation" (GO: 0010584) (Figure 6A), and the KEGG pathway analysis showed that these DEGs were preferentially clustered in "phenylpropanoid biosynthesis" (ko00940) and "phenylalanine metabolism" (ko00360) (Figure 6B and Table S4). Phenylpropane synthesis is involved in the formation of sporopollenin [31]. As shown in Figure 6C, two key genes (*PAL1* (*Bra005221* ($\log_2$FC: $-1.588$)) and *PAL4* (*Bra029831* ($\log_2$FC: $-2.376$))) encoding phenylalanine ammonia-lyases (PAL) were obviously down-regulated, and in the downstream process of hydroxyphenyl lignin synthesis, the genes encoding 4-coumarate-CoA ligase (*4CL1* (*Bra030429* ($\log_2$FC: $-1.348$)) and *4CL5* (*Bra001819* ($\log_2$FC: $-1.287$); *Bra001820* ($\log_2$FC: $-1.038$)) and cinnamyl alcohol dehydrogenase (*CAD2* (*Bra031216* ($\log_2$FC: $-2.334$)) and *CAD5* (*Bra011510* ($\log_2$FC: $-2.499$)))) were also down-regulated; however, the gene encoding cinnamyl-CoA reductase gene (*CCR2*) was up-regulated [32]. The changes in the expression levels of these genes probably affected phenylpropane synthesis, thus affecting the accumulation of lignin and sporopollenin, which would modulate the pollen wall composition and eventually affect pollen viability.

### 2.5. qRT-PCR Validation

The expression of several representative DEGs related to pollen development was validated by qRT-PCR. The results show that the selected genes measured by qRT-PCR had a similar expression pattern to those detected by RNA-seq analysis (Figure 7 and Table S5). The key genes in glutathione metabolism pathway (*GPX7*, *PGD1* and *G6PD2*) were up-regulated in the same trend as the RNA-seq results. Additionally, the key genes in phenylpropane metabolic pathway (*PAL1*, *PAL4*, *4CL1*, *4CL5*, *CAD2* and *CAD5*) were down-regulated by qRT-PCR in accordance with RNA-seq results, suggesting that these key genes played an important role in this process. In addition, the actin family of genes (*ACT1*, *ACT3*, *ACT4* and *ACT12*) showed the same results. The selected gene *Bra006228* (*EXO70C2*), associated with pollen growth, showed higher expression in the maintainer line (231–330) than the Ogura CMS line (Tyms). The gene (*Bra016891* (*SPL9*)) related to anther development showed up-regulation in the Ogura CMS line (Tyms). *Bra016891* and a gene related to carbohydrate metabolism (*Bra004835* (*BGLU15*)), which showed remarkably higher expression in the Ogura CMS than maintainer line (231–330), represent putative candidates for controlling pollen development in Ogura CMS Chinese cabbage.

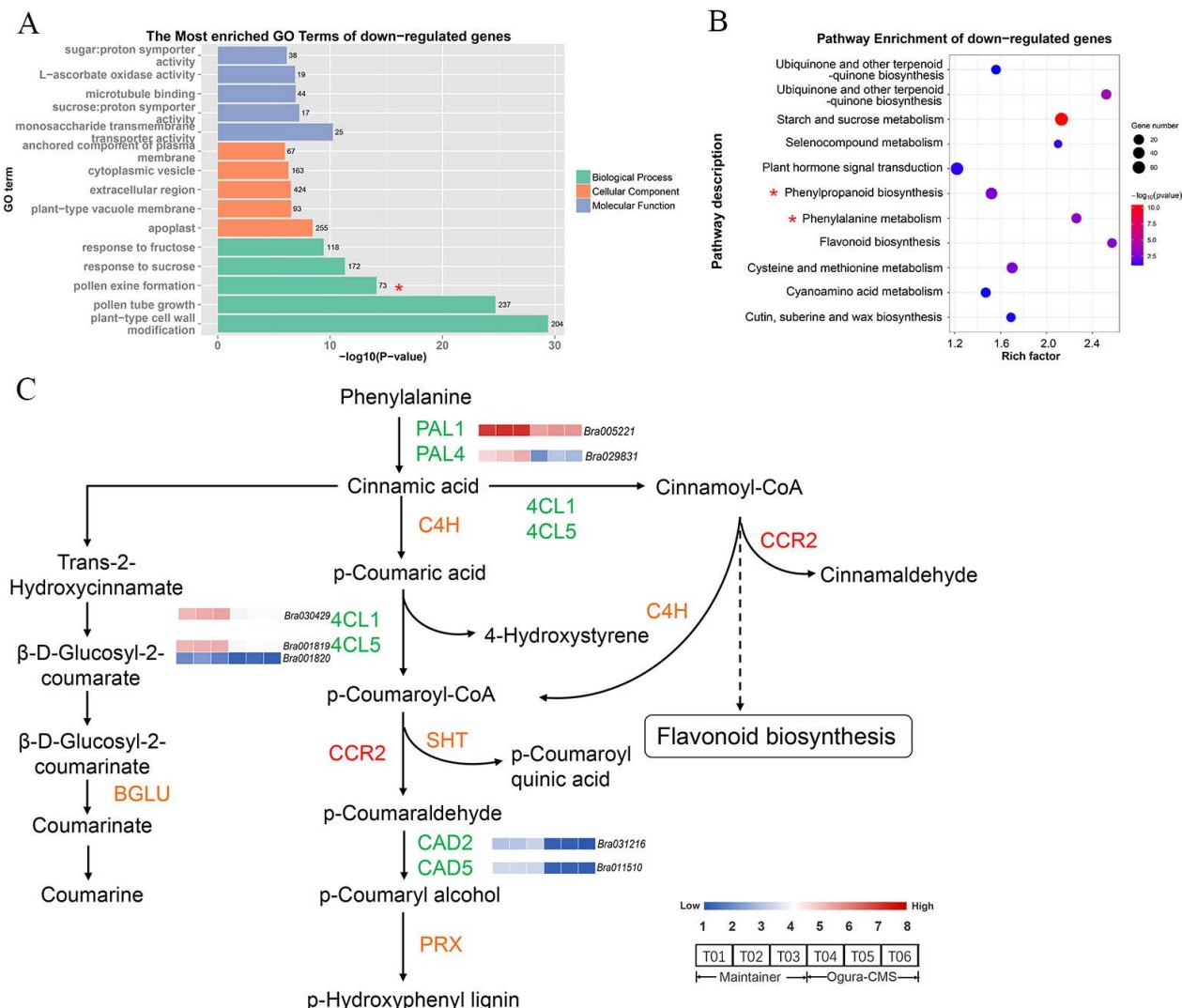

**Figure 6.** Enrichment analysis of genes down-regulated in the maintainer line vs. the Ogura CMS line. (**A**) GO analysis of down-regulated DEGs; (**B**) KEGG enrichment analysis about pollen exine formation for down-regulated DEGs; (**C**) pathway of phenylpropane biosynthesis (ko00940). PAL1, phenylalanine ammonia-lyase 1; PAL4, phenylalanine ammonia-lyase 4; C4H, cinnamate 4-hydroxylase; 4CL, 4-coumarate-CoA ligase; SHT, spermidine hydroxycinnamoyl transferase; CAD5, cinnamyl alcohol dehydrogenase 5; PRX, peroxidase; CCR2, cinnamyl-CoA reductase; BGLU, β-glucosidase. The red is the up-regulated enzyme gene; the orange represents the enzyme gene with no change in expression; the green represents the down-regulated enzyme gene. T01, T02 and T03 represented 3 libraries (the maintainer line, 3 times replication); T04, T05 and T06 represented 3 libraries (the Ogura CMS line, 3 times replication).

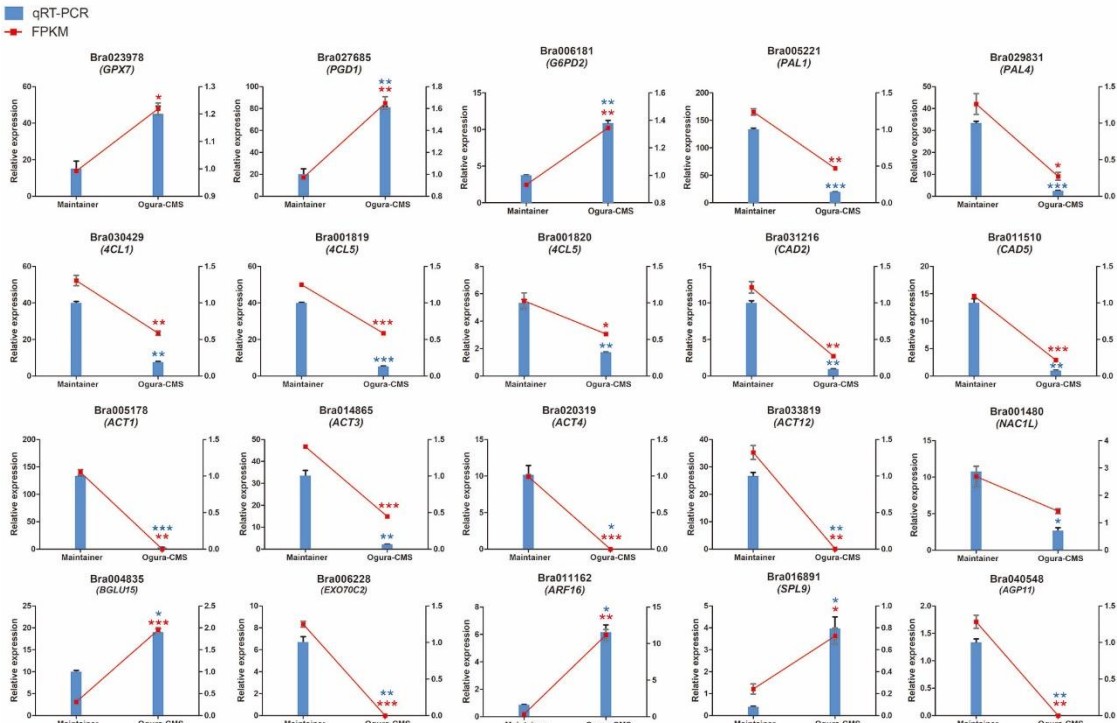

**Figure 7.** qRT-PCR validation of selected DEGs. The blue columns represent qRT-PCR results and the red lines represent RNA-seq results. *GPX7*: glutathione peroxidase 7; *PGD1*: glucose 6-phosphate dehydrogenase 1; *G6PD2*: glucose 6-phosphate dehydrogenase 2; *PAL1*: phenylalanine ammonia-lyases 1; *PAL4*: phenylalanine ammonia-lyases 4; *4CL1*: 4-coumarate-CoA ligase 1; *4CL5*: 4-coumarate-CoA ligase 5; *CAD2*: cinnamyl alcohol dehydrogenase 2; *CAD5*: cinnamyl alcohol dehydrogenase 5; *ACT1*: actin 1; *ACT3*: actin 3; *ACT4*: actin 4; *ACT12*: actin 12; *NAC1L*: NAC1-Like; *BGLU15*: beta glucosidase 15; *EXO70C2*: exocyst subunit Exo70 family protein C2; *ARF16*: auxin response factor 16; *SPL9*: squamosa promoter binding protein-like 9; *AGP11*: arabinogalactan protein 11. Student's *t*-test was used for statistical analysis of data from the two lines (* $p < 0.05$; ** $p < 0.01$; *** $p < 0.001$).

## 3. Discussion

### 3.1. The Accumulation of ROS May Lead to the Premature Degradation of Tapetum, Which Affects the Formation of Pollen Exine

As the outermost layer of anther parietal cells, tapetal cells can provide nutrients such as protein, fat and carbohydrate for the development of microspore by secreting vesicles and self-degradation [33]. Studies have shown that the premature or delayed degradation of tapetum will affect pollen development and lead to pollen abortion [34].

The timely degradation of the tapetum by programmed cell death (PCD) is crucial for microspore development and pollen wall maturation, and reactive oxygen species (ROS) have been shown to be involved in programmed tapetal cell death in plants [35]. Studies have confirmed that mitochondrial protein WA352 can interact with COX11, inhibiting the ROS clearance function of COX11, leading to the increase of ROS, and inducing the occurrence of tapetum PCD in advance, thus leading to male sterility in rice [36]. In addition, tapetum also secretes sporopollenin precursors, which are involved in the construction of the outer wall of pollen [37]. In the way of programmed cell death, the decomposed substances of tapetum are filled into the pollen wall to form a layer of pollen wall composed of lipids, proteins and pigments [29]. The function of tapetum and the formation of pollen envelope are the important processes of pollen development after meiosis.

In our study, *GPX6*, *GPX7*, *G6PD2* and *PGD1* genes were up-regulated in the process of glutathione metabolism, suggesting a large accumulation of ROS in the Ogura CMS lines. The down-regulation of PAL, 4CL and CAD enzymes in the phenylpropane metabolic

pathway also suggests the abnormal synthesis of sporopollenin, which may cause the pollen to be less effective in resisting external damage.

In conclusion, according to previous studies, excessive ROS may lead to early degradation of tapetum and affect the uptake of nutrients by pollens. Meanwhile, the abnormal development of tapetum may further hinders the formation of sporopollenin in the pollen exine, which eventually leads to pollen abortion. However, further studies are needed to determine their progressive relationship.

### 3.2. Cytoskeletal Actin Dynamics Were Probably Involved in Pollen Sterility

Actin is an inclusion in the cytoskeleton that contains a microtubule system and a microfilament system. Limited actin content in the cytoskeleton or disorders of the cytoskeleton system can cause pollen abortion [15]. The protein interaction network depicted in Figure S3 shows the relationship of cytoskeletal genes from the KOG enrichment analysis and the thickness of the straight lines is proportional to the strength of correlation.

We identified *ACT1* (*Bra017166*, *Bra000010* and *Bra005178*) and *ACT3* (*Bra007025* and *Bra014865*), which are expressed in mature pollen [38]. *ACT4* (*Bra006719* and *Bra020319*) and *ACT12* (*Bra018210*, *Bra019486* and *Bra033819*) belong to a reproductive actin subclass predominantly expressed in developing reproductive tissues, such as pollen, pollen tubes, ovules and developing seeds, and are expressed at very low levels in vegetative organs [25]. The plant actin bundler *PLIM2s* have been shown to regulate actin bundling in different cells [39]. We found three genes that encode pollen-specific LIM proteins—*PLIM2A* (*Bra000404*, *Bra004939* and *Bra039313*), *PLIM2B* (*Bra032650* and *Bra033233*) and *PLIM2C* (*Bra014447*)—in the cytoskeleton category of the KOG enrichment analysis. The complete suppression of the *PLIM2s* completely disrupted pollen development, producing abortive pollen grains in transgenic *Arabidopsis thaliana*, whereas the partial suppression of the *PLIM2s* arrested pollen tube growth, resulting in short and swollen pollen tubes [40]. Our transcriptome sequencing results also seem to suggest that one of the causes of pollen abortion in Ogura CMS lines is the down-regulation of *PLIM2s*.

The actin depolymerizing factor (ADF) plays a key role in actin remodeling, which can increase the actin filaments treadmilling rate [26]. Studies have monitored the expression and subcellular localization of ADF7 and ADF10 proteins during male gametophyte development, pollen germination and pollen tube growth. ADF7 is related to the development of the microspore nucleus and vegetative nucleus of mature grains at the stage of low metabolic activity, and ADF10 is associated with the actin filaments in the process of gametophyte development, particularly with the arrays surrounding the apertures of the mature pollen grains [41]. In the study, the down-regulation of *ADF7* (*Bra010456*, *Bra013901* and *Bra019151*) and *ADF10* (*Bra028304* and *Bra029119*) between the CMS and maintainer lines, which seems to confirm the above results. Another gene, *CAP1*, was found to be involved in cyclase-related proteins, which are important regulators of actin turnover. However, its exact function in regulating actin polymerization, and especially its contribution to actin nucleotide exchange activity, is still not fully understood. In *Arabidopsis*, homozygous *cap1* alleles resulted in short stature and reduced pollen germination efficiency [42]. The *CAP1* gene is located at the core of the protein interaction network; therefore, its expression, even in small quantities, may greatly affect pollen tube growth. DEGs associated with actin, which affect the dynamics of the cytoskeleton by regulating actin motion and play a key role in pollen development, could be explored in connection to the mechanism of male sterility.

## 4. Materials and Methods

### 4.1. Morphological Observation

At the blooming stage, flowers of both the Ogura CMS and maintainer lines were collected. The observation of inflorescences was carried out under a Nikon SMZ645 stereoscopic microscope (Nikon Corporation, Tokyo, Japan). For the ultrastructure observation by electron microscopy, the anthers or pollen grains from the newly opened flowers were

evenly tiled on the sample table attached to the conductive adhesive and sprayed with an ion-sputtering instrument. Then, a Hitachi SU3500 scanning electron microscope was used to observe and take photos.

### 4.2. Observation of Meiotic Chromosomal Behaviors

Young inflorescences were collected and fixed overnight in Carnoy's solution (ethanol: glacial acetic acid, 3:1 *v/v*) at room temperature, and then stored at 4 °C in 70% ethanol for later use. The anthers were removed from the inflorescences with a dissecting needle under a stereomicroscope and soaked in a mixture of cellulase (0.5% *w/v*) and pectinase (0.5% *w/v*) to dissolve the microspore cell wall, and then incubated in a citric acid buffer at 37 °C for 4 h. The prepared slides were stained with PI solution (40 g/mg) for 5 min, and then observed under a fluorescence microscope.

### 4.3. DAPI Staining

The stamens were collected from flower buds of Chinese cabbage at different developmental stages fixed with Carnoy's fixative (ethanol: glacial acetic acid; 3:1). The fixed buds were used for observing different developmental stages of microspores under a fluorescence microscope and photographed after staining with 4,6-diamidino-2-phenylindole (DAPI). A few drops of DAPI dye were added to the glass slides, and the nuclei were stained for 10 min and thereafter flushed with running tap water. Excess water was removed from the slides using filter paper, after which a drop of fluorescent sealing was added and observed under a fluorescence microscope at 360–400 nm wavelengths.

### 4.4. Transcriptome Sequencing and Identification of DEGs

The stamens of the isogenic Ogura CMS and maintainer lines with different cytoplasmic backgrounds—named as Tyms and 231–330, respectively—of Chinese cabbage were used in this study. The plants were cultivated in experimental plots at the Henan Academy of Agricultural Sciences (Yuanyang, China). Fifty flower buds were trimmed from 10 different plants during microsporogenesis (five buds from each plant), and pooled samples were kept at −80 °C after snap-freezing in liquid nitrogen prior to RNA extraction and sequencing.

A total amount of 1.5 µg RNA per sample was used as input material for rRNA removal using the Ribo-Zero rRNA Removal Kit (Epicentre, Madison, WI, USA). The cDNA library was sequenced via sequencing by synthesis (SBS) technology using Illumina HiSeq 2500 high-throughput Sequencing platform. Raw data (raw reads) of fastq format were firstly processed through in-house perl scripts. In this step, clean data (clean reads) were obtained by removing reads containing adapter, reads containing ploy-N and low quality reads from raw data. Then, the clean data were aligned to the *Brassica rapa* reference genome, v1.5 (http://brassicadb.org/brad/, accessed on 7 August 2019) and the comparative efficiency ranged from 72.73% to 90.22%. StringTie (1.3.1) was used to calculate FPKMs of both mRNAs and coding genes in each sample. Gene FPKMs were computed by summing the FPKMs of transcripts in each gene group.

Differential expression was determined from repeat count data using EdgeR (1.10.1), a Bioconductor software package [43]. Over-dispersion across transcripts was reduced following Poisson's model as well as the empirical Bayesian method for improving the reliability of analysis. A fold change (FC) ≥2 and false discovery rate (FDR) <0.05 were used as criteria for screening the DEGs. All the analyses were performed using software tools on BMK Cloud (https://international.biocloud.net/zh/software/tools/, accessed on 7 August 2019).

### 4.5. Annotation and Functional Analysis of DEGs

The annotation information for the new gene was obtained through the alignment of DEG sequences against the nonredundant (Nr), BLAST search [44], Swissport [45], Gene Ontology (GO) [46], Clusters of Orthologous Group (COG) [47] and Kyoto Encyclopedia

of Genes and Genomes (KEGG) [48] databases. GO enrichment analysis was carried out for the genes with inter-sample differences, whereas Cluster Profiler [49] was used for the genes of biological process, molecular function and cell components. A hypergeometric test in enrichment analysis was performed to find GO terms that were significantly enriched compared to the entire genomic background. The terms obtained from the enrichment results were visualized using the ggplot2 package in R. The euKaryotic Orthologous Group (KOG) diagrams were drawn by using BMKCloud software tools (https://international.biocloud.net/zh/software/tools/, accessed on 7 August 2019).

### 4.6. Quantitative Real-Time PCR (qRT-PCR) Validation

In order to verify the accuracy of the DEG data obtained by RNA-seq, the relative expression levels of the 14 key DEGs and 6 genes selected randomly were analyzed by qRT-PCR. RNA was extracted from 100 mg anthers of Tyms and 231–330 and reverse-transcribed into cDNA as a template for qRT-PCR. The fluorescence quantitative primers for the selected DEGs and the housekeeping gene *GAPDH* (internal control) were designed on NCBI (https://www.ncbi.nlm.nih.gov/, accessed on 8 June 2021) and are listed in Table S6. The specificity and amplification efficiency of the primers were assessed by qRT-PCR. A LightCycler 480 II System (Roche, Basel, Switzerland) and SYBR Premix Ex TaqTM (TaKaRa, Dalian, China) were used for qPCR. The relative expression was calculated according to the $2^{-\Delta\Delta Ct}$ method [50]. A 20 µL reaction-mixture was used in the qPCR; the PCR mixture contained 10 µL of SYBR Premix Ex TaqTM (Tli RNaseH Plus), 0.8 µL of 10 mM concentrations of the forward and reverse primers, 2.0 µL of cDNA (30 ng/µL) and 6.4 µL of dH$_2$O. Each PCR was repeated three times as technical replicates. The qPCR was performed using the following profile: initial denaturation at 95 °C for 5 min, and then 45 cycles including denaturation for 10 s at 95 °C, annealing for 10 s at 58 °C and extension for 15 s at 72 °C. The results for relative expression according to the qRT-PCR and RNA sequencing were visualized using GraphPad prism 5 software.

### 5. Conclusions

In this study, cytological observations showed that the anthers that were about to blossom were cavitary and the nucleus gradually degrades in the late uninuclear stage of microspore development until it completely disappears, which causes the sterility rate to reach 100%. Using high throughput RNA-seq technology, a total of 8052 genes were identified as differentially expressed between the maintainer (231–330) and Ogura CMS Chinese cabbage lines. We identified key DEGs clustered in the glutathione oxidation pathway and phenylpropane synthesis pathway, which are probably involved in pollen abortion in the Ogura CMS Chinese cabbage line, due to the accumulation of ROS and abnormal outer wall composition in pollens. In addition, the DEGs related to cytoskeleton and energy metabolism were also enriched, which indicated a possible role in regulating post-meiotic progression during pollen development. The putative candidate DEGs reported in this study pave the way for the exploration of regulatory mechanisms in controlling sterility and fertility in Ogura CMS and its maintainer lines during microsporogenesis.

**Supplementary Materials:** The following are available online at https://www.mdpi.com/article/10.3390/horticulturae7060157/s1, Figure S1. Morphological features of flower organs and inflorescence in the Ogura CMS line and its maintainer line of Chinese cabbages at anthesis stage. (A–H) a floret at anthesis stage with normal flower organs in the maintainer line; (a–h) a floret at anthesis stage with shorter filaments and anthers without pollen in the Ogura CMS line. Bar = 5 mm. Figure S2. Observation of chromosome behaviors at meiosis of Ogura CMS line and maintainer line in Chinese cabbage. (A–E) chromosome behaviors at meiosis in maintainer line (231–330); (E–H) chromosome behaviors at meiosis in sterile line (Tyms). A, F: Diakinesis; B, G: Metaphase I; C, H: Telophase I; D, I: Metaphase II; E, J: Telophase II; Bar = 10 µm. Figure S3. A protein interaction network was constructed for genes related to cytoskeleton. the thickness of the straight lines is proportionate to the strength of correlation. Table S1. Statistical analysis of morphological properties of maintainers and Ogura sterile lines. The following data are based on the average of 30 samples. Table S2. All the

DEGs were classified into 25 categories by using the KOG database. Table S3. All DEGs' expression level in clusters of "Cytoskeleton". Table S4. The expression levels of all down-regulated genes and their annotations. Table S5. The expression values of qRT-PCR and transcriptome sequencing. Table S6. List of primer sequences for quantitative fluorescence verification involving selected DEGs and internal reference.

**Author Contributions:** X.W. and F.W. conceived and designed the experiments; Y.Z., S.Y. and Z.W. performed the experiments; G.S., X.Z. and Z.X. analysis the data; L.H., R.L. and B.T. prepared the figures and tables; X.W., Y.Y. and U.K.N. drafted and revised the manuscript critically. All authors have read and agreed to the published version of the manuscript.

**Funding:** This work was supported by the Zhongyuan Scholar Program (202101510003), National Natural Science Foundation of China (31801874), Programs for Science and Technology Development of Henan Province (212102110124) and Sci-Tech Innovation Team of Henan Academy Agricultural Sciences (2021TD06).

**Institutional Review Board Statement:** Not applicable.

**Informed Consent Statement:** Not applicable.

**Data Availability Statement:** Data is available at NCBI SRA accession PRJNA657160. The data presented in this study are available upon request from the corresponding author.

**Acknowledgments:** We would like to express our thanks to the anonymous reviewers for their useful comments.

**Conflicts of Interest:** The authors declared no conflict of interest and absence of any commercial or financial benefits of this research.

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
