# Peer review of "Comparative Transcriptome Identifies Gene Expression Networks Regulating Developmental Pollen Abortion in Ogura Cytoplasmic Male Sterility in Chinese Cabbage (Brassica rapa ssp. pekinensis)"

_horticulturae, doi:10.3390/horticulturae7060157_

Round 1

Reviewer 1 Report

The authors aimed to identify a network of key genes involved in developmental pollen abortion in Ogura CMS Chinese cabbage by transcriptome and RT-qPCR analyses. The work done made it possible to reveal the relationship between the expression of a group of genes involved in cytoskeleton system, glutathione redox reaction and phenylpropane biosynthesis pathway, and pollen abortion in Chinese cabbage.

This research has many valuable and meaningful results. However, there are several issues that need the authors attention:

Line 47-48: “CMS is determined by the mitochondrial genome and associated with pollen sterility phenotype that can be suppressed or counteracted by nuclear genes known as fertility restorer genes [6]" – I think this sentence should to the next paragraph, which was about Brassica CMS genetic background.

Line 82: Do the authors mean do you mean “Chinese cabbage” instead of “rapeseed”?

Line 81-85: Please provide references to support your points.

Line 96: Hav the authors done any statistical analyses on the size of flowers organs? It would be more persuasive if the authors provided the differences in the flower size between the CMS and maintainer lines.

Line 97: Please provide a legend for Figure S1.

Line 100: Please provide a legend Figure S2.

Figure 1 and Figure S1: In Figure 1, the size of the anthers in the maintainer line were bigger than in CMS line. However, the CMS line' anthers were bigger than that of the maintainer line in Figure S1c, f and h. Could the authors elaborate on these differences?

Line 131: Please provide actual number of DEGs for the “carbohydrate transport and metabolism” category.

Line 139-143: Please delete “etc.” and list all the ACTIN genes that were downregulated. I would highly suggest the authors to provide fold change in gene expression of ACTINs and VLN5 to make your points clearer. The authors should also provide the FPKM read-count of the DEGs in the supplement data.

Section 2.2: I would highly recommend the authors to extend your report on cytoskeleton system related genes because their roles are equally important to that of GSH oxidation genes and sporopollenin related genes.

Line 165: Do the authors mean “GPX” instead of “DPX”?

Figure 5: Please provide explanation for T01 – T06.

Line 184-189: Please again support your claims with analysed data (e.g. no. of DEG in each category, fold change in gene expression). The authors should also provide the read-count of all down-regulated DEGs in the supplement data.

Line 188: Please elaborate on why CCR2 was up-regulated when the availability of its substrates, the cinnamic acid, p-coumaroyl-CoA and cinnamoyl CoA, were reduced stemming from the down-regulation of PALs and 4CL? Please also provide references to support your points.

Figure 5: Please explain the use of colour to code enzymes (orange, red and green) in the phenylpropane pathway. Please provide explanation for T01 – T06.

Line 197: Please remove letter “P” in front of phenylalanine ammonia-lyase 4.

Section 2.3 and 2.4: Have the authors conducted any analyses to determine the level of H2O2 derived ROS species and lignin content in the CMS and maintainer lines? I believe those data would strongly support your hypothesized mechanisms of male sterility in Chinese cabbage.

Line 202: Have the authors validated the expression level of genes related to glutathione redox reaction (GPXs, PGD1 and G6PD2), lignin synthesis (PALs and 4CLs) and cytoskeleton system (ACTs)? I would recommend the author to present those data along with the 8 randomly selected genes to make a stronger conclusion.

Line 201-211: Please again provide analysis data to support your claims of higher/lower expression levels in the main text. Please also add gene acronyms to the main text since they were used in Figure 7.

Figure 7: Have the authors applied any statistical analyses on the RT-qPCR derived data and read-counts? Please present the results on the graphs and mention the method used in the Figure legend. Please also provide the full form of acronyms listed in Figure 7.

Section 3.1: This discussion section was poorly written. More importantly, discussing about mitochondrial mutations was not highly relevant to the main aim of this study which was focus on identification of a gene network regulating pollen abortion by transcriptome analysis.

Line 232: Please list the involved genes in the main text instead of vaguely mentioning them as “some key enzymes”.

Line 242: Please provide a figure legend for Figure S3.

Line 257-259: Please paraphrase this sentence to convey clearer message.

Line 259: The definition of ADF acronym must move forward to Line 257. Please check and use a consistent definition for ADF in Line 141 and 259.

Line 260: Do the authors mean “tread milling rate” instead of “trade milling rate”?

Line 289: Do the authors mean “cellulase (0.5 % w/v)” instead of “cellulose”?

Line 309: Please provide more technical details regarding the high-throughput sequencing and data analysis, such as Illumina sequencer model, sequencing depth and reference genome assembly version. All tools (edgeR, R) and packages should also be provided with their version to ensure better reproducibility.

Line 330: RT-qPCR validation: please provide detailed information regarding tissue samples used for RT-qPCR.

Line 333: Could the authors confirm the internal control used in this study, ACTIN or GAPDH as listed in Table S2? If ACTIN was used as an internal control, would it compromise the RT-qPCR derived data since ACTs were reported to differently express between 2 lines in this study?

Line 344:  Please also provide the number of biological replicates used in RT-qPCR analyses. It is important to have both biological and technical replicates in a single run to eliminate data biases.

Line 360: Please check the SRA data on NCBI. SRA accession PRJNA657160 was not found.

All the Figures were packed with information however they are small in size and poor in quality making them really difficult to read.

Reviewer 2 Report

Dear Authors,

I have reviewed the manuscript "Comparative Transcriptome identifies Gene Expression Networks Regulating Developmental Pollen Abortion in Ogura- Cytoplasmic Male Sterile of Chinese Cabbage (Brassica rapa ssp. pekinensis)" that describes the cytological and transcriptome differences between Ogura and the maintainer line. The introduction is complete and the results are well described even if could be improved. The discussion is full of grammar errors and not clear in explaining and discussing what found in the results. I invite the authors to completely revise the discussion. In Materials and Methods is missing the description of reads mapping after Illumina sequencing. Please add this part.

My final recommendation is to reconsider after major revision due to the weak of the discussion and the missing of an experimental procedure.

Here  below you can find minor comments:

Lines 218-220: could you please revise this sentence: “In this CMS system, mitochondrial ORF138 controlled the sterility [7, 32], pollen production failed due to destroying the synergistic relationship

among various substances [36].” being more specific briefly indicating which substances are involved.

Line 234: please, change “To sum up” with a synonymous word.

Line 238: pleae, revise the sentence: ”Actin is an inclusion in the cytoskeleton contains microtubule system and microfilament system.”

Line 244: chenge “which expressed” with “which are expressed”

Lines 240-242: I do not understand the meaning or scope of this sentence: “The protein interaction network shows the relationship between cytoskeletal related genes from KOG enrichment analysis and the thickness of the straight lines is proportionate to the strength of correlation (Figure S3).” it seems more likely a result description or a figure lengend.

Line 249: please, replace “functions” with “encode” in the sentence “We found three genes that functions pollen-specific LIM proteins.

Line 254: delete “was”

Lines 255-256: the relative expression of PLIM2s demonstrates or suggests?

Lines 257-259: please revise this sentence “In addition, the expression level of genes showed significant differences in ADF7 (Bra010456, Bra013901 and Bra019151), ADF10 (Bra028304 and Bra029119) and CAP1 (Bra034629) between sterile and fertile lines.”. I would write: “In addition, the expression level of ADF7 (Bra010456, Bra013901 and Bra019151), ADF10 (Bra028304 and Bra029119) and CAP1 (Bra034629) showed significant differences between sterile and fertile lines.”

Line 267: please revise “found to involve”

Line 302: Paragraph 4.4. Transcriptome sequencing and identification of DEGs. In this parafraph is missing how the sequence reads were used before EdgeR as this package works on read counts and not raw data.

Line 311: please, delete “and produced a large amount of high-quality raw data.”

Kind regards.

Round 2

Reviewer 1 Report

The authors have carefully responded to all questions from the reviews and made the necessary changes to the manuscript. However, there are suggestions for the authors and some minor problems need to be addressed.

The authors need to provide all the supplementary tables and cite them correctly in the manuscript.

I highly suggest to integrate the fold change in expression level data in the manuscript to show how significant the assessed genes were up- or down- regulated.

Line 205 (follows up with the authors’ response #14): Thank you for providing a reference to support your point. It is clear from the provided reference that the CCR is capable of catalysing 2 different substrates located on 2 branches of the lignin biosynthesis pathway. However, I am more curious on your point of “the lack of substrate led to the up-regulation of CCR2 gene”. To my understanding, plant generally suppress the expression of a certain enzyme encoding genes (i.e. less amount of enzyme is produced) in response to the low substrate concentration. With the down regulation of PAL, it is obvious that the availability of all the downstream products/substrates, including those for 4CL and CCR, were reduced. Could the authors please kindly elaborate on how the lack of substrate led to the up-regulation of CCR2 gene?

Quality of Figures: Please kindly check the quality of the Figures. The small texts are very difficult to read even after zoomed in. Please also make sure that the process of word-to-pdf conversion did not reduce their resolutions.

Reviewer 2 Report

Dear Authors,

I have reviewed the revised manuscript. The most important point in my previous review was the weakness of the conclusions paragraph as it was not clear and not analyzing the results. In this second version, the conclusions are improved even if a bit short. In particular the first paragraph 3.1.  

Despite this aspect, my final advice is to accept in the present form.

Kind regards

Author Response

Dear Reviewer,

Thanks very much for your valuable advice for improvements of our work, which makes the article more rigorous and perfect. We have tried our best to improve our work in the revised version according to your good suggestions. Thank you again!

Kind regards